# Smooth Muscle Cell Phenotypic Switch Induced by Traditional Cigarette Smoke Condensate: A Holistic Overview

**DOI:** 10.3390/ijms24076431

**Published:** 2023-03-29

**Authors:** Laura Bianchi, Isabella Damiani, Silvia Castiglioni, Alfonso Carleo, Rossana De Salvo, Clara Rossi, Alberto Corsini, Stefano Bellosta

**Affiliations:** 1Section of Functional Proteomics, Department of Life Sciences, University of Siena, Via A. Moro 2, 53100 Siena, Italy; 2Department of Pharmacological and Biomolecular Sciences “Rodolfo Paoletti”, Università degli Studi di Milano, Via Balzaretti 9, 20133 Milan, Italy; 3Department of Pulmonology, Hannover Medical School, Carl-Neuberg-Straße 1, 30625 Hannover, Germany; 4IRCCS MultiMedica, 20099 Sesto San Giovanni, Italy

**Keywords:** atherosclerosis, cigarette smoke, KLF4, MYOCD, EIF2AK2/PKR, phenotypic switch, smooth muscle cells

## Abstract

Cigarette smoke (CS) is a risk factor for inflammatory diseases, such as atherosclerosis. CS condensate (CSC) contains lipophilic components that may represent a systemic cardiac risk factor. To better understand CSC effects, we incubated mouse and human aortic smooth muscle cells (SMCs) with CSC. We evaluated specific markers for contractile [i.e., actin, aortic smooth muscle (*ACTA2*), calponin-1 (*CNN1*), the Kruppel-like factor 4 (*KLF4*), and myocardin (*MYOCD*) genes] and inflammatory [i.e., IL-1β, and *IL-6*, *IL-8*, and galectin-3 (*LGALS-3*) genes] phenotypes. CSC increased the expression of inflammatory markers and reduced the contractile ones in both cell types, with *KLF4* modulating the SMC phenotypic switch. Next, we performed a mass spectrometry-based differential proteomic approach on human SMCs and could show 11 proteins were significantly affected by exposition to CSC (FC ≥ 2.7, *p* ≤ 0.05). These proteins are active in signaling pathways related to expression of pro-inflammatory cytokines and IFN, inflammasome assembly and activation, cytoskeleton regulation and SMC contraction, mitochondrial integrity and cellular response to oxidative stress, proteostasis control via ubiquitination, and cell proliferation and epithelial-to-mesenchymal transition. Through specific bioinformatics resources, we showed their tight functional correlation in a close interaction niche mainly orchestrated by the interferon-induced double-stranded RNA-activated protein kinase (alternative name: protein kinase RNA-activated; PKR) (EIF2AK2/PKR). Finally, by combining gene expression and protein abundance data we obtained a hybrid network showing reciprocal integration of the CSC-deregulated factors and indicating KLF4 and PKR as the most relevant factors.

## 1. Introduction

Atherosclerosis is the leading cause of myocardial infarction and stroke and is the major cause of death in the Western world. The exact cause of atherosclerosis is still uncertain, but certain traits, conditions, or habits may behave as risk factors for the disease. Inflammation and hypercholesterolemia are hallmarks and potent promoters of cardiovascular disease (CVD), in addition to other well-defined risk factors that contribute to the multifactorial processes involved in disease progression (diabetes, hypertension, aging, and smoking) [1]. Among the environmental factors that may contribute to cardiovascular risk, incidence, and severity, cigarette smoking is one of the biggest threats to current and future world health [2]. Tobacco smoke interacts with inflammatory cytokines to produce endothelial dysfunction [3] and induces pro-inflammatory and pro-atherosclerotic effects in vascular tissue [4].

Smooth muscle cells (SMCs) are present in the media of human arteries, and are considered protective against atherosclerotic plaque destabilization [5]. Contractile SMCs are the most prominent cell type in the healthy vessel wall [6]. Normally, they are elongated, with a spindle-shaped morphology, and produce a well-characterized set of contractile markers, including smooth muscle actin (ACTA2), smooth muscle myosin heavy chain (MYH11), smooth muscle protein 22-alpha (SM22α/TAGLN), smoothelin (SMTN), and calponin (CNN1) [7]. However, SMCs are not terminally differentiated, and retain the ability to undergo a phenotypic switch from a contractile to a dedifferentiated synthetic state and express inflammatory markers and a phagocytic activity in response to environmental cues [8]. Cholesterol, a known risk factor for atherosclerosis, induces a phenotypic switch in SMCs [9] which, like monocytes and macrophages, can take up and store excess lipids and form foam cells [9,10,11]. Upon cholesterol loading in vitro, mouse SMCs reduce the expression of typical smooth muscle markers, including *ACTA2*, α-tropomyosin, and *CNN1* genes and increase the expression of macrophage-related ones, such as *CD68*, galectin-3 (*LGALS-3*), and the membrane protein transporter ATP-binding cassette A1 (*ABCA1*) [11]. These phenotypic changes lead to the downregulation of the miR-143/145–myocardin axis in a Kruppel-like factor 4 (KLF4-dependent manner [9,11]. KLF4 is a transcription factor implicated in SMC proliferation and dedifferentiation, and it plays a fundamental role in modulating cell pluripotency [7].

Cigarette smoking is a leading cause of mortality and morbidity, and cigarette smoke (CS) is another risk factor for inflammation-related disorders, such as atherosclerosis [2]. The CS condensate (CSC) from the particulate phase of the CS aerosol contains lipophilic components that may pass the respiratory membranes and reach the blood stream, thus representing a cardiac and vessel-systemic risk factor. It has been shown that CSC may affect cerebral SMCs through myocardin (MYOCD) and KLF4-dependent mechanisms [12].

The aim of our study was to profoundly characterize the effects of CSC on aortic SMC phenotypic switch from a biochemical, transcriptional, and differential proteomic approach. Our data show functional processing of CSC-deregulated proteins, detected using liquid chromatography–tandem mass spectrometry (LC–MS/MS) and bioinformatics; and through western blot/immunocytochemistry analyses we identified the interferon-induced double-stranded RNA-activated protein kinase (alternative name: protein kinase RNA-activated; PKR) (EIF2AK2/PKR; henceforth PKR) as the main relevant factor involved in SMC phenotypic plasticity. Combination of gene expression and differential proteomics data using MetaCore led to the generation of a hybrid network where all experimental factors were highly integrated under the direct control of KLF4. In addition, the gene/protein hybrid network evidenced a tight functional cross-talk between PKR and KLF4; hence suggesting that vascular SMC phenotypic plasticity induced by CSC is orchestrated by PKR and KLF4.

## 2. Results and Discussion

### 2.1. Effect of Cigarette Smoke Condensate on Murine SMC Phenotypic Switch

We previously demonstrated phenotypic changes in mouse SMCs loaded with free cholesterol complexed to methyl-β-cyclodextrin [11]. This resulted in less differentiated cells that lacked SMC markers but showed an increased inflammatory profile. They actually had an enhanced expression of *Lgals-3*, interleukins, and of the cholesterol transporters *Abca1* and *Abcg1*, which are associated with increased cell proliferation and migration, as well as with synthesis of extracellular matrix and related proteases, such as matrix metalloproteinases (MMPs) [11]. These are specific features that may promote atherosclerosis [7,11,13].

With the aim of understanding CSC effects on SMC plasticity, we incubated mouse SMCs with CSC lipophilic components (see Section 3) for 48 h. Then, we evaluated the expression of specific markers of both contractile (*Acta2* and *Cnn1*) and inflammatory (*Lgals-3, Abca1, Abcg1*) phenotypes. As shown in Figure 1a, after incubating mouse SMCs with CSC we observed a significant decrease in *Cnn1* (up to 50%, *p* < 0.005 vs. control), and *Acta2* (up to 70%, *p* < 0.001 vs. control) mRNA levels, which was confirmed using western blot analysis. In parallel, we observed a statistically significant increase in the expression of *Lgals-3*, both at mRNA and protein levels (up to 3-fold and 5-fold, respectively, *p* < 0.001, Figure 1b), and of *Abca1* and *Abcg1* mRNA (up to 4-fold and 2-fold, respectively, *p* < 0.01).

We and others have previously shown that SMC cholesterol loading converts SMCs to a foam cell-like state by downregulating the miR-145–MYOCD axis [9,11]. MYOCD is a potent myogenic transcriptional coactivator that controls the expression of *ACTA2* and *CNN1* and negatively regulates SMC inflammatory activation and vascular disease, and its levels are reduced during atherosclerosis in association with SMC phenotypic changes [7,14,15]. As shown in Figure 2, the addition of CSC significantly reduced *Myocd* expression in SMCs (up to 70% reduction, *p* < 0.005) reproducing the pathological situation observed during atherogenesis [15]. As expected, the expression of *Klf4,* a known repressor of *Myocd* [16], was increased (doubled, *p* < 0.001) by treatment with CSC. MYOCD is required for phenotypic transition of cultured SMCs in response to PDGF [17]. It also regulates SMC transition toward an inflammatory phenotype [18] and induces the expression of miR-145, which is one of the most important miRNAs in CVD according to its high expression in SMCs [9]. MiR-145 knockdown induces atherogenesis in mice regardless of hypercholesterolemia; its expression decreases with plaque progression and its overexpression reduces atherosclerosis [19,20,21]. MiR-145 regulates SMC function in intimal hyperplasia, inhibits SMC proliferation and migration, and may regulate *MYOCD* expression and SMC phenotypic switching [7,22,23]. The incubation with CSC reduced miR-145 expression by 40% (*p* < 0.01 vs. control, Figure 2), in agreement with the data obtained by incubating murine SMCs with cholesterol [9,11,24].

### 2.2. Effect of Cigarette Smoke Condensate on Human SMC Phenotypic Switch

Next, we confirmed the murine data in human cells. Human aortic SMCs (HSMCs) were incubated for 48 h with CSC (30 μg/mL) and then HSMC phenotypic switch was evaluated. The addition of CSC significantly reduced the SMC-specific markers *ACTA2* and *CNN1* (by 40% and 60%, respectively, *p* < 0.001 vs. control, Figure 3a), and increased the expression of inflammation-related markers, such as *LGALS-3, CD68, IL-6,* and *IL-8* (by 80%, 70%, more than 4-fold, and 7-fold, respectively, Figure 3b). *IL-1β* mRNA levels increased four times and this was also confirmed at the protein level by both western blot analysis (3-fold increase, *p* < 0.005) and by confocal microscopy (Figure 3c). As expected, this phenotypic switch was consequent to a stimulation of *KLF4* expression by almost five times. Although less markedly than in mouse, also *MYOCD* expression was reduced by CSC exposure (Figure 3d).

To further asses the induction of a phenotypic switch by CSC, we measured its effect on HSMC proliferation using cell counting. As shown in Figure 4, the addition of CSC stimulated HSMC proliferation and the effect was evident after 48 h of incubation.

Subsequently, we measured the capacity of CSC to stimulate HSMC migration, another pro-atherogenic feature associated with SMC phenotypic switch. To test the effect of CSC, we used the wound healing assay (or conditional migration assay) that measures the ability of HSMCs to migrate and reclose a lesion induced on the cell monolayer. The addition of CSC markedly increased HSMC reclosure rate (Figure 5). In fact, 88% of the lesion area was healed after 24 h of incubation with the CSC, versus only 60% of the lesion observed in control HSMCs (Figure 5).

### 2.3. CSC Effects on VSMC Proteomic Profile and Combined Functional Analyses of Proteins and Genes Significantly Deregulated by the CSC Treatment

To further our understanding on the biochemical basis of the CSC-induced phenotypic switch, we performed an MS-based differential proteomic analysis in HSMCs. Eleven proteins were significantly affected by the treatment with CSC (FC ≥ 2.7, *p* ≤ 0.05, Table 1).

Only two of them were up-regulated, i.e., pyrroline-5-carboxylate reductase 2 (PYCR2) and PKR (EIF2AK2 in the heatmap). All the others were downregulated, as shown by the heatmap in Figure 6. In particular, the vertical dendrogram, on the left of Figure 6, evidences the clustering of the differing proteins into three groups according to their differential abundance in the two tested conditions.

Cluster A includes PYCR2 and PKR. On the contrary, cluster B2 contains the most downregulated proteins in CSC-exposed cells, i.e., protein MON2 homolog (MON2), atrial natriuretic peptide receptor 3 (NPR3), and collectin-12 (COLEC12). The other six proteins, less downregulated by CSC treatment, are grouped in cluster B1 and they are: serine/arginine-rich splicing factor 5 (SRSF5), transmembrane protein 43 (TMEM43), transmembrane emp24 domain-containing protein 1 (TMED1), (E3-independent) E2 ubiquitin-conjugating enzyme (UBE2O), schlafen family member 5 (SLFN5), and elongator complex protein 2 (ELP2).

Interestingly, identified proteins are differentially active in several signaling pathways related to pro-inflammatory cytokine and IFN expression, to inflammasome assembly and activation, cytoskeleton regulation and SMC contraction, mitochondrial integrity and cellular response to oxidative stress, proteostasis control via ubiquitination, and cell proliferation and epithelial-to-mesenchymal transition.

We built hybrid networks to delineate the CSC-affected pathways and to evaluate how the aberrant abundance/expression of protein/gene differences may affect SMCs and tissue physiology in atherosclerosis onset, as well as to integrate data from classical biochemical analyses and those from the MS-based proteomic approach. In the MetaCore suite, we co-processed WB- or PCR-detected factors deregulated in CSC-treated HSMCs and MS-identified differences. The functional processing of these proteins/genes evidenced their tight functional correlation in a close interaction niche, which corroborates their involvement in specific CSC-affected pathways and highlights their relevance as biomarkers of CSC exposure.

#### 2.3.1. Protein Hybrid Network

Firstly, we obtained a shortest path network (SPN) of deregulated proteins by including in the processed list, along with the 11 MS-identified differences, also IL-1β, whose CSC-induced upregulation was proved by RT-PCR, WB, and immunocytochemistry (Figure 3c). Despite the very few not-experimental factors added by the software to cross-link experimental proteins that were not directly related, all the protein differences were included into the net, except for TMEM43. This proves the tight functional correlation existing among them and strongly suggests that they may play critical roles in the phenotypic switch we observed. Since several of these deregulated proteins were not previously associated with cellular response to CS, their functional cross-talk may offer a new perspective on CSC effects on SMCs, and they could be evaluated as biomarkers and targets in SMC transdifferentiation. PKR, IL-1β, UBE2O, SLFN5, and SRSF5 (alternative name: pre-mRNA-splicing factor SRP40; SRP40) became the central hubs of the protein SPN as they established the highest number of interactions (Figure 7). In particular, the kinase/adapter protein PKR was the most relevant one, being cross-linked to the highest number of net nodes.

PKR is implied in a plethora of cellular functions spanning from signal transduction and apoptosis, to cell proliferation and differentiation, by modulating p53/TP53, PPP2R5A, ILF3, and IRS1 activities [25,26,27,28], and by regulating various signaling pathways, such as p38 mitogen-activated protein kinase (p38 MAPK), NF-kB, and insulin signaling pathways, as well as of transcription factors, e.g., JUK, STAT1, STAT3, IRF1, and ATF3, involved in gene expression of pro-inflammatory cytokines and IFNs [29,30]. Although studies on PKR in SMCs have only recently intensified [31], several of PKR’s known activities underline a close and multilevel correlation between the increase in its expression and the biochemical and phenotypic changes we described in HSMC after CSC treatment.

PKR is an innate immune/inflammatory-cytokine-associated protein kinase, one of the four kinases composing the integrated stress response (ISR) system. The ISR is involved in cellular adaptation to stress; and its kinases, when activated, cause an immediate gene expression reprogramming by phosphorylating the α subunit of eukaryotic translation initiation factor 2 (eIF2α) [32], a member of the PERK–eIF2α–ATF4 pathway that is involved in SMC transdifferentiation and vascular calcification [33]. PKR is activated or induced by different types of cellular stresses, including viral infection, inflammatory signals, and oxidative, metabolic, mechanical, and endoplasmic reticulum (ER) stresses [27,30,31,34]. CSC is known to cause oxidative stress [35,36,37,38], and cigarette smoke provokes ER stress and inadequate protein turnover in alveolar epithelial cells [39]. Reasonably, in CSC-exposed cells, oxidative and ER stresses may initiate the up-regulation of PKR that, for its part, could trigger inflammatory signaling through the above-mentioned pathways and nuclear factors, thus auto-supporting its expression. In fact, PKR is induced by pro-inflammatory cytokines, e.g., TNF-α, IL-1, INF-γ, and, depending on the cell type, PKR itself induces the release of the pro-inflammatory IL-18, IL-1β, and high mobility group box 1 (HMGB1) alarmin proteins [40]. PKR is actually reported to interact with several components of the macrophage inflammasome, regulating its activity, and, finally, in induction of pyroptosis [30,40].

PKR is also implied in metabolic syndrome and insulin resistance [41,42]. Furthermore, PKR inhibition attenuates inflammation, oxidative stress, and apoptosis marker gene expression in SMCs incubated in high fructose (HF) medium [31]. Interestingly, HF causes proliferation and phenotypic switch of these cells [31]. As a matter of fact, although further analyses are needed, the dedifferentiation of SMCs increased their proliferation and migratory capability, and the acquisition of an inflammatory state triggered by CSC exposure may be orchestrated by PKR, as similarly reported for other vessel stressors.

Vascular SMCs are characterized by a phenotypic plasticity that allows them to adapt in presence of environmental changes and during disease development [8]. PKR is gaining relevance in this context, not only as a regulator of transdifferentiation but also by controlling extracellular matrix degradation and remodeling. Activated PKR actually mediates the increase in matrix-metalloproteinase 2 and 9 (MMP2 and MMP9, respectively) gene expression and protein activation [43], which may obviously facilitate the CSC-induced SMC migration.

The signal transducer activator of transcription 3 (STAT3) is a transcription factor induced by PKR that is used by MetaCore to cross-link PKR to IL-1β and to ELP2, also known as STAT3-interacting protein 1 (STATIP1 in MetaCore). In addition to its role in the expression of pro-inflammatory cytokines, STAT3 promotes cell survival and proliferation [44], and is involved in phenotypic switch of synthetic SMCs. Its overexpression actually inhibits MYOCD-induced up-regulation of contractile phenotype-specific genes in SMCs [45]. STATIP1 modulates the ligand-dependent activation of STAT3 and its overexpression blocks IL-6-dependent STAT3 activation in vitro [46]. Since the IL-6/STAT3 pathway can modulate SMC proliferation, migration, and expression of MMPs [47], reduced abundance of STATIP1 is a biochemical observation that properly fits with the behavior changes induced by exposing HSMC to CSC, as we described above. IL-6 is associated with an increased cardiovascular risk [48], and induces senescence-associated calcification of SMCs by activating the STAT3/p53/p21 signalling pathway [49]. Therefore, the CSC stimulation in SMCs may even be related to in vivo atherosclerotic lesion calcification by inducing IL-6 upregulation and reducing STATIP1 occurrence.

Interferon regulatory factor 1 (IRF1) is another critical net-point added by the software for experimental hub cross-linking. It controls PKR, IL-1β, SLFN5, and NPR3 gene expression and is modulated by SRP40, as shown in Figure 8. IRF1 directly regulates the expression of inflammation and migration-related genes in a human microglial cell line [50] and induces *IL-6* and *IL-1β* transcription [51]. These data suggest a possible correlation of IRF1 in controlling the HSMC inflammatory state induced by CSC, by modulating the above-listed experimental proteins affected by CSC and by functionally correlating all of them with its own activity. NPR3 is the transmembrane receptor of the C-type natriuretic peptide (CNP) and exerts an anti-proliferative, anti-migratory, and anti-inflammatory role [52]. NPR3 signaling impedes cardiac and vascular remodeling by suppressing SMC proliferation and collagen deposition [53]. Interestingly, CNP-KO mice suffer from endothelial dysfunction, hypertension, and atherosclerosis onset [54]. Therefore, the evident downregulation of NPR3 (Figure 6) plausibly contributes to the phenotypic switch induced by CSC in HSMCs by negatively affecting pathways that counteract cellular proliferation and ECM fiber synthesis. Additionally, SLFN5 is an IRF1-controlled factor downregulated by the CSC treatment. It is involved in the inhibition of endothelial mesenchymal transition (EMT) and of E-cadherin-repression, by downregulating the zinc finger E-box-binding homeobox 1 [55] as well as in the suppression of MMP expression and of cellular proliferation, migration, and invasiveness in different types of cancer [56,57,58,59,60]. The downregulation of SRP40 induced by CSC treatment may concur with SLFN5 in facilitating HSMC transdifferentiation and migration. SRP40 knockdown reduces the expression of tight-junction proteins increasing blood–tumor-barrier permeability [61]. In addition, SRP40 modulates the alternative splicing of the glucocorticoid receptor (GCR) and, consequently, defects in SRP40 activity may affect GCR signaling in reason of GRα/β ratio variations [62]. Glucocorticoids are crucial in maintaining cardiovascular health and have been described to influence the development of atheromatous plaques [63]. CSC-induced abundance reduction in SRP40 may hence have in vivo deleterious effects by dysregulating GCR signaling.

Noteworthy, three CSC-downregulated proteins are under the direct control of GCR, and two of them, MON2 and COLEC12 (Figure 7), are from the B2 cluster of the dendrogram shown in Figure 6. This means they are among the most downregulated proteins of the analysis. MON2 is a Golgi apparatus protein taking part to an evolutionarily conserved endosome-associated membrane remodeling complex active in the endosome-to-Golgi transport pathway, which is an integral part in autophagy-mediated longevity [64,65]. SMC senescence and apoptosis occurring in mature plaques increase plaque vulnerability, stenosis, medial degeneration, and thrombogenicity by converting the initial fatty streaks to a fibro-atheroma lesion [66]. In addition, apoptotic SMC remnants result in nucleating centers of calcium deposition and plaque calcification, thus further increasing the possibilities of plaque rupture [66]. On one hand, autophagy is active in the stress response of SMCs and plays a pivotal role in determining their phenotypic switch under growth factor stimulation, e.g., PDGF; on the other hand, it reduces foam cell formation, lipid accumulation, and lesion mineralization by regulating apoptosis [66]. Indeed, the downregulation of MON2 we observed in response to CSC may be of particular interest in SMCs of stressed vessels from classical cigarette smokers. The lipophilic condensate components of CS may, in fact, affect the vessels, by reducing MON2 presence and vesicle protein cargo trafficking and recycling, thus negatively impacting SMC survival and atherosclerotic lesion development. Nonetheless, MON2 activity is also involved in Wntless rescue from lysosomal degradation and, consequently, it may interfere with Wnt secretion [67]. Despite its signaling promoting SMC survival [68], Wnt exerts several, although debated, functions in cardiovascular physiopathology and its downregulation may reduce plaque instability [69]. The role of MON2 in SMC transdifferentiation and in atherosclerosis has not yet been investigated and, even if its dysregulation may apparently have antithetic effects, its control on vesicle cargo, protein recycling, and autophagy, along with its consistent CSC-dependent downregulation, make this protein an interesting novel biomarker of stressed SMCs that undoubtedly deserves to be further investigated.

The transmembrane scavenger receptor (SR) C-type lectin COLEC12 (CL-P1), which is up-regulated by hypoxia, is involved in ox-LDL binding and internalization processes, regardless of intracellular cholesterol content [70]. COLEC12 is principally present in cells of placental, stromal, and macrophage origin, and plays a role in cell-to-cell adhesion, similarly to selectins [71]. Despite SRs exerting a relevant role in cardiovascular diseases [72,73], COLEC12 is under-investigated in vessels and its detection in vascular endothelial cells [74] is controversial [75]. COLEC12 is consistently reduced in SMCs exposed to CSC and we may suppose that such a decline could diminish the SMC reciprocal interaction in vitro, thus facilitating cell migration.

Finally, UBE2O is a member of the E2 family of the ubiquitin-proteasome system (UPS) that acts as an E2/E3 hybrid enzyme and that is principally expressed in heart and skeletal muscle. UPS is crucial in cellular proteostasis and critical for several cellular functions, such as gene transcription, inflammatory response, endocytosis, intracellular protein trafficking, and angiogenesis, by modulating relative abundance of ubiquitinated proteins [76]. In addition, differential ubiquitination changes properties, their reciprocal interactions and localization of proteins, hence profoundly impacting on their function and on pathways in which they work [76]. Since it is also endowed with a self-contained quality-control activity for substrate recognition [77], UBE2O downregulation may deeply affect protein dynamics and cellular functions in SMCs treated with CSC. Namely, its depletion enhances the tumor necrosis factor (TNF)-associated factor 6-(TRAF6)/NF-kB signaling [78] that has been recently correlated with anoikis resistance and cell spreading in cancer [79], and whose inappropriate activation causes uncontrolled innate immune responses [78]. Interestingly, induced expression of UBE2O suppresses IL-1β/TRAF6-induced signaling by inhibiting the polyubiquitination of E3 ligase TRAF6 [78]. The *IL-6*-enhanced expression we observed leads us to suppose that reduced presence of UBE2O results in an upregulation, maybe via TRAF6, of NF-kB with consequent intensification of the IL-6/NF-kB signaling, critical in vessel inflammation [80].

Reduced abundance of UBE2O may hence be associated with the inflammatory state induced by CSC in HSMCs, as well as with a generalized variation in proteoform pattern(s) that may participate in the phenotypic switch.

#### 2.3.2. Protein/Gene Hybrid Network

We have demonstrated that CSC induces HSMC phenotypic switch via the KLF4/MYOCD axis. The protein/gene hybrid SPN, built by using the entire list of factors deregulated at protein or transcript levels in CSC-treated cells, was centred on KLF4, which actually turned out to be the main central hub (Figure 8). This SMC-plasticity regulator directly correlates with all the CSC-deregulated genes except for CNN1, and with all the deregulated proteins except for the SLFN5 (see Appendix A). Nonetheless, 34 and 21 of the interactors MetaCore added to CNN1 and SLFN5, respectively, are under the direct functional control of KLF4. Since all these interactors converge on CNN1 or SLFN5, KLF4 reasonably exerts a tight control on both of them.

Although under-studied, SLFN5 knock-down is involved in EMT of breast cancer cells [81]. Consequently, the indirect inhibitory control exerted by KLF4 on SLFN5 may result in a depletion of its protein product, as we observed using MS analysis, with consequences in the contractile-to-mesenchymal switch. This is perfectly aligned with the KLF4 downregulation of CNN1.

Differentiated vascular SMCs do not normally express *KLF4* in vivo, but they transiently induce its expression after vascular injury [82]. *KLF4* induction was recently reviewed as crucial in the initial dedifferentiation of SMCs to the mesenchymal-like phenotype, which may allow, depending on external stimuli, further molecular changes toward the other four known SMC phenotypes [7,83]. The (i) fibroblast-like, (ii) macrophage-like, (iii) osteogenic-like, and (iv) adipocyte-like phenotypes acquired by SMCs after dedifferentiation may profoundly impact vessel dysfunctions and atherosclerosis onset and development [7,84]. Reasonably, *KLF4* induction due to CSC exposure, may trigger the SMC phenotypic changes we described above. According to the hybrid SPN, KLF4 evidently orchestrates the cellular response to the treatment and its interactors and related pathways may offer a new perspective on CS effects on the cardiovascular system.

In addition, it is of great relevance that the two main actors that our data delineated on the stage of SMC transdifferentiation, i.e., KLF4 and PKR, not only directly interact but they also indirectly correlate through six proteins known to, or suspected to, exert relevant roles in SMC phenotypic switch, proliferation, migration, and inflammatory state development. These are the above described STAT3 and IRF1, inhibitor of nuclear factor kappa-B kinase subunit alpha (CHUK; IKK-α in the MetaCore SPN), SUMO-conjugating enzyme UBC9 (UBE2I; E2I in the MetaCore SPN), basic helix–loop–helix ARNT-like protein 1 (BMAL1), and transcription factor Sp1/Sp3 complex (Appendix A). Noteworthy, STAT3, positively regulated by both PKR and KLF4, induces the *LGALS-3* gene expression [85] that widely participates in vascular SMC transdifferentiation (vide infra). While PKR induces NF-kB signaling by activating IKK-α [86], KLF4 is among the stemness-related genes that are directly induced by IKK-α via an interaction with the aryl hydrocarbon receptor (AhR) [87], which has recently been recognised as a major player in CVDs [88]. The balance between sumoylation and desumoylation controls the differentiation of adult stem cells and KLF4 is directly involved in this process [89].

Interestingly, the circadian clock transcription factor BMAL1, which is essential for normal circadian variations in SMC contraction, was described to promote a phenotypic switch of SMCs towards fibroblast-like cells and to stabilize atherosclerotic plaques [90]. As shown in Appendix A SPN, KLF4 and PKR are both under BMAL1 control [91]. Since this transcription factor also suppresses vascular SMC migration, deregulation of BMAL1 may affect SMC behavior by modulating PKR and KLF4 activity.

Finally, the induction of both PKR and KLF4 by Sp1 again stresses the relevance they have in the onset and development of atherosclerotic pathology. Sp1 is actually involved in the main events of atherosclerosis development, such as vascular SMC proliferation, inflammation, lipid metabolism, plaque stability, and endothelial dysfunction [92]. PKR and KLF4 may also act as Sp1 effectors in modulating SMC behavior. Sp1, cooperatively with Sp3, in fact mediates basal expression of PKR in the absence of IFN stimulation [93] and induces *KLF4* in phenotypically modulated SMCs [17].

Unlike STAT3, SP1, BMAL1, IKK-α, UBE2I, and IRF1 are factors that have received a marginal interest in vascular injury and related atherosclerotic events. Nevertheless, their close functional correlation with both PKR and KLF4 suggests that these proteins could exert relevant roles in vascular SMC phenotypic variations, or at least in those induced by CSC exposure.

Contractile SMCs are regarded as differentiated and quiescent cells under physiological conditions, expressing a panel of typical contractile proteins crucial in maintaining vascular tension. In particular, a correlation between the loss of the contractile phenotype of SMCs exposed to CSC and the activity of KLF4 is suggested by the direct or indirect inhibitory interactions that this transcription factor establishes with markers of contraction—markers we proved to be down-regulated, such as ACTA2 (included in the MetaCore actin node) and CNN1, and not to mention the inhibitory effect KLF4 has against MYOCD, the master regulator of smooth muscle-specific gene expression (Figure 8).

Evidently, CSC unleashes a stress condition in SMCs whose molecular effectors/effects consistently overlap with those triggered by other vessel injury events that induce SMCs to reduce the expression of contractile phenotypic markers and migration inhibitors and to acquire a proliferative and migratory behaviour.

SMC alterations are responsible for the transition of these cells from a contractile phenotype to an active synthetic one able to release paracrine mediators, including TNF-α and IL-6 [94]. This function promotes the further synthesis of cytokines, e.g., IL-8 and IL-1β [95], and growth factors that sustain ECM remodeling, vascular SMC proliferation, and migration [96]. In particular, the transient mesenchymal-like phenotype triggered by *KLF4* expression is characterized by an LGALS-3 positive state [97], whose expression is directly induced by KLF4 [98,99]. *LGALS-3* inhibition increases *CNN1* and *ACTA2* gene expression in human pulmonary arterial SMCs under hypoxic conditions [100]. LGALS-3 is considered a predictive marker for the development and progression of CVDs [101], including atherosclerosis [102]. In addition to promoting SMC migration and phenotypic switching to the synthetic type through the Wnt/β-catenin signalling pathway [103], LGALS-3 is actually intimately involved in acute inflammation and in its chronicization [104,105,106].

The detrimental osteogenic and proinflammatory phenotypes derive from the LGALS-3-positive SMC transitional state [99] as well as from the macrophage-like state. Macrophage-like SMCs, whose phenotypic conversion is facilitated by high cholesterol levels and ox-LDL via *KLF4* induction [9,11,107], express LGAL-3 and other macrophage marker genes coding for F4/80 (ADGRE1), CD11b (*ITGAM*), CD68, CD45 (*PTPRC/CD45*), and CD116 (*CSF2RA*) antigens [108]. Among them, the *CD68* gene codes for macrosialin, a plasma membrane glycoprotein involved in phagocytic activities of tissue macrophages as well as in macrophage homing by binding lectin and selectin and allowing cell crawling over selectin-presenting substrates. Interestingly, Allahverdian et al. showed that ~40% of all CD68-positive macrophages within human coronary artery lesions are derived from SMCs [109]. As CSC induces *CD68* expression in SMCs in vivo, we can suppose that the lipophilic components of CSC not only induce vascular SMC switch to the mesenchymal-like phenotype but that it may even facilitate its further switch to the macrophage-like one.

Among the CSC-deregulated proteins that interact with KLF4, PYCR2 deserves attention. It is an essential enzyme in proline biosynthesis and promotes cancer proliferation and progression [110,111]. Proline-abundance increase may lead to E-cadherin reduction in the plasma membrane [112], a process previously associated with ox-LDL treatment of SMCs, which induces SMC proliferation and disassembling of their adherens junctions [113]. Accordingly, increased PYCR2 levels may support SMC proliferation and migration.

Conversely to its fate in the hybrid protein SPN (Figure 8), the MS-detected protein-difference TMEM43 is included in the protein/gene hybrid SPN and its function is under the direct control of KLF4 (Appendix A). TMEM43 is a structural protein of the inner nuclear membrane highly and uniformly expressed in fibroblasts and vascular SMCs [114] and whose mutations cause fatal arrhythmia in humans [115,116,117,118,119]. It interacts with the lamins A/C, B1, emerin, and SUN domain-containing protein 2 (SUN2) and is probably involved in emerin localization [115,116,117]. As an interactor of the linker of nucleoskeleton and cytoskeleton (LINC) complex, it may therefore act in mechanosignaling and in related regulation of gene expression, cell signaling, nuclear structure, and chromatin architecture. In addition, TMEM43 is a widespread cytoplasmic plaque protein of the zonula adherens from various epithelial cell types [120]. We may consequently speculate that the decreased abundance of TMEM43 could lead to gene expression reprogramming as well as to a reduced cell–cell interaction in vascular SMCs exposed to CSC, with consequent phenotypic switch to a mesenchymal-like one.

In line with TMEM43 and PYCR2, KLF4 also downregulates SRP40 [121], which causes as described above, dysregulation of tight junction proteins, and COLEC12, which is involved in cell-to-cell adhesion (vide supra) [122]. In addition, KLF4 indirectly controls vesicular trafficking of proteins by modulating gene expression of two other proteins that we detected to be downregulated by CSC treatment: *TMED1* and *MON2* [122]. According to this latter gene modulation, KLF4 may further affect autophagy, plaque stability, and mineralization. Lesion development, as well as SMC proliferation and migration, may be conditioned by KLF4 also downregulating *STATIP1* and NPR3 expression [122,123]. Finally, KLF4 role in CSC-related inflammation may pass through *UBE2O* regulation [123].

In addition to its effects on classical markers of SMC differentiation and inflammatory state (i.e., ACTA2, MYOCD, IL-1β, IL-6, IL-8, LGALS3), KLF4 activities properly correlate with the deregulation in protein abundance we observed in SMCs exposed to CSC. This stresses the functional relevance of the MS/MS delineated biomarker panel in SMC behavioral changes induced by CSC, despite a number of our protein biomarkers not having been effectively studied yet in SMC phenotypic switch.

## 3. Materials and Methods

### 3.1. Cell Culture

Murine smooth muscle cells (SMCs) were isolated from the intimal–medial layer of aortae of C57BL/6 mice of both sexes (The Jackson Lab, Bar Harbor, ME, USA) as described in [11]. Subconfluent SMCs were incubated in DMEM (Euroclone, Milan, Italy) supplemented with 0.2% essential-fatty-acid-free albumin.

Human aortic SMCs (PCS-100-012, ATCC, MA, USA) were cultured in ATCC Vascular Cell Basal Medium (PCS-100-030, ATCC; 500 mL supplemented with 500 µL ascorbic acid, 500 µL rh EGF, 500 µL rh insulin and rh FGF-b, 25 mL glutamine), 5% FBS (ATCC Vascular Smooth Muscle Growth kit), and 5 mL penicillin–streptomycin 100× (Euroclone, Milan, Italy).

### 3.2. Cigarette Smoke Condensate

Cigarette smoke condensate (CSC) was kindly provided by British American Tobacco (Southampton, UK). CSC contains the lipophilic components present in both the gas and the particulate phases of a standard traditional cigarette (1R6F, manufactured and provided by the University of Kentucky, Lexington, KY, USA). The use of standardized reference cigarettes provides better uniformity of experimental responses both within the same laboratory and also between laboratories [124]. CSC was dissolved in dimethylsulphoxide (DMSO) to yield a final concentration of 24 mg/mL. Final maximal DMSO concentration in all samples was adjusted to 0.1% [125].

### 3.3. Confocal Microscopy

Cells were washed with PBS and fixed in 4% paraformaldehyde for 15 min, permeabilized by adding 0.1% Triton and subsequently saturated with 5% BSA for 1 h. After adding primary antibodies in blocking buffer, cells were incubated overnight in the dark, washed, and stained with the secondary anti-bodies Alexa Fluor 488 and 546 (Thermo Fisher Scientific, Monza, Italy). Cells were then washed with PBS and stained with DAPI. Images were acquired using a confocal microscope (FRET FLIM, 40× objective lens, Leica Microsystems, Milan, Italy).

### 3.4. RNA Isolation and Reverse Transcription

Total RNA from cells was extracted with the Direct-zol^TM^ RNA MiniPrep Plus kit (Zymo Research, Irvine, CA, USA). Concentration and purity of RNA were measured using a Nanodrop 1000 spectrophotometer (Thermo Fisher Scientific). A total of 1 μg of total RNA was reverse transcribed using the iScript gDNA Clear cDNA Synthesis kit (1725035, Bio-Rad, Milan, Italy), according to manufacturer’s instructions.

### 3.5. Quantitative RT-PCR

Quantitative RT-PCR was performed by using iTaq Universal SYBR Green Supermix and specific primers for selected genes [24]. Mouse primer sequences used for qPCR analysis are shown in Table 2. The analyses were performed with the CFX CONNECT TM Real Time System (BioRad). PCR cycling conditions were as follows: 95 °C for 1 min, 40 cycles at 95 °C for 10 s, and 60 °C for 30 s. The results were analyzed using the ΔΔCt method using the expression values of the reference gene GAPDH. The fold-change was calculated using 2^−(ddCt)^, comparing control cells versus CSC-treated cells.

### 3.6. PCR Arrays

The mRNA expression in human SMCs was measured with a human muscle contraction PCR array (Bio-Rad) and an IL-1β signaling pathway PCR array (Bio-Rad) as indicated by the manufacturer. Briefly, the amplified cDNA was diluted with nuclease-free water and added to the SsoAdvanced Universal SYBR^®^ Green Supermix. A volume of 20 μL of the experimental cocktail was added to each well of the array. Real-Time PCR was performed on a CFX CONNECT TM Real Time System (Bio-Rad) with the following thermal profile: activation—1 cycle, 95 °C for 2 min; denaturation—40 cycles, 95 °C for 5 s; annealing/extension—40 cycles, 60 °C for 30 s; melt curve—65–95 °C (0.5 °C increments) 5 s/step. All data from the PCR were analyzed using CFX Maestro software v 2.3 (Bio-Rad).

### 3.7. miRNA Expression

A total of 20 ng of extracted RNA was reverse transcribed into cDNA using the miRCURY LNA RT Kit (Qiagen, Hilden, Germany). Mature miRNA expression levels were measured using the miRCURY LNA SYBR Green PCR Kit (Qiagen). Results from qPCR were normalized using miR-103A and relative gene expression was quantified with the ∆∆Ct method.

### 3.8. Protein Isolation, Quantification, SDS-PAGE and Western Blot

For the preparation of total cell lysates, cells were washed with ice-cold PBS and lysed with lysis buffer (NaCl 150 mM, TRIS 50 mM pH 7.6, NONIDET P-40 0.5%, and protease inhibitors (Merck, Milan, Italy)). Protein concentration was determined using a Pierce BCA Protein Assay kit (Pierce, Rockford, IL, USA) and samples were run on SDS-PAGE. The different proteins were detected using specific primary antibodies: ACTA-2 (ab7817, 1:300), LGALS-3 (ab76245, 1:5000), and IL-1β (ab2105, 1:200) were from Abcam (Cambridge, UK); tubulin (T6199, 1:1500) from Sigma-Aldrich (Milan, Italy). Quantification was performed using densitometric analysis using Image Studio Lite software v. 3.1 from Li-Cor Bioscience (Lincoln, NE, USA).

### 3.9. Cell Proliferation

Cells were seeded in 24-well plates at a density of 3 × 10^4^ cells/well. After 24 h, cells were incubated with medium containing 0.4% FBS to synchronize cells in the G0 phase of the cell cycle. After 72 h, cells in control dishes were counted with a Coulter Counter (Beckman Coulter, Life Scientific, Milan, Italy) and this was considered the “basal” number of cells at T0. Then, medium was removed and replaced with medium containing 30 μg/mL CSC and 10% of FBS for 24 and 48 h. Cell number was measured and compared to the zero time-point [126].

### 3.10. In Vitro Directional Migration (Wound Healing Assay)

Cells were plated in 24-well plates and grown to confluence. Cell monolayers were scratched with a 200 μL pipet tip in a straight line. Thereafter, the cell monolayer was washed with growth medium to remove detached cells. Cells were then incubated with medium containing 0.4% FBS and CSC (30 μg/mL). Images of the wounded area were acquired at the same spot at different time-points using an inverted microscope (Axiovert 200; 10× objective lens, Carl Zeiss, Milan, Italy) equipped with a digital camera. Quantification of the wound area was performed using ImageJ, and cell migration was expressed as a percentage of wound area at different time-points compared to initial wound area (T0) [127].

### 3.11. Comparative Mass Spectrometry Proteomics

Relative quantitative mass spectrometry (MS) was performed using label-free quantification (LFQ). Protein samples were reduced, alkylated, and then digested in 5 mM Dithiothreitol (DTT, at 55 °C for 30 min), 15 mM 2-Iodoacetamide (IAA, at room temperature for 20 min), and 0.1 µg/µL trypsin (overnight at 37 °C), respectively. Protein samples were purified through C18 reverse phase column Zip-tip purification, concentrated in a speedvac vacuum concentrator, and dissolved in 4 µL 0.1% *v*/*v* formic acid solution. Samples were analysed in triplicates using nano-liquid chromatography–high resolution mass spectrometry (nLC–HRMS) on a Dionex Ultimate 3000 nano-LC system (Sunnyvale, CA, USA) connected to an Orbitrap Fusion™ Tribrid™ mass spectrometer (Thermo Scientific), equipped with a nano-electrospray ion source. In particular, peptide mixtures were pre-concentrated onto an Acclaim PepMap 100—100 µm × 2 cm C18 (Thermo Scientific)—and separated on an EASY-Spray column ES802A, 25 cm × 75 µm ID packed with Thermo Scientific Acclaim PepMap RSLC C18, 3 µm, 100 Å. The peptides were eluted with a gradient from 96% buffer W (0.1% formic acid in water) to 95% buffer A (0.1% formic acid in 75% acetonitrile) at a constant temperature (35°C) and flow rate (300 nL/min) for 144 min. MS spectra were collected in data-dependent mode over an m/z range of 375–1500 Da at 120,000 resolutions, and a cycle time of 3 s between master scans. Higher-energy collision dissociation (HCD) was performed with collision energy set at 35 eV and positive polarity. MS raw data were processed using MaxQuant v2.2 [https://www.maxquant.org, (accessed on 2 July 2022)] [128]. In particular, the settings to identify and quantify the proteins were: human taxonomy (UP000005640.fasta), trypsin digestion, cysteine carbamidomethylation (as fixed modification), methionine oxidation, and N-term acetylation or Met-loss (as dynamic modifications). Only the proteins identified by two or more unique peptides, presenting Q-values lower than 0.05, and observed in 4 of 6 replicates at least in one condition were considered in the analysis. The LFQ values from technical replicates were averaged and then normalized using DEP R-package v.1.20.0 [July 2022] [129]. The statistical comparison between CSC and control experimental groups was performed using v.3.54.1 [July 2022] [130] and the differentially abundant proteins were represented in a heatmap plot. In particular, heatmap clustering was obtained according to Ward’s method of the Euclidean distances.

### 3.12. Functional Analysis of Detected Differences in the CSC vs. Control Comparison

Functional analysis of acquired data was performed using MetaCore v21.3 (Clarivate Analytics, Boston, MA, USA) integrated software suite for functional analysis of experimental data. MetaCore consists of a manually annotated database of human protein–protein, protein–DNA, and protein–compound interactions, metabolic and signaling pathways, and the effects of bioactive molecules both in healthy and disease status from scientific literature.

Accession numbers of MS-identified proteins and factors found as dysregulated from WB or PCR analysis were imported into MetaCore and co-processed using the “shortest path” algorithm (SPA), set to “high trust interaction”.

SPA permits to correlate experimental factors with proteins not present in the submitted list but supported by the MetaCore database to functionally correlate experimental proteins that do not directly interact. We allowed 2 steps for experimental protein cross-linking and avoided canonical pathways. Nets were built limiting protein processes to individual proteins and excluding their involvement in multimeric complexes.

Generated pathway maps were prioritized according to their statistical significance (*p* < 0.001) and networks were graphically visualized as nodes and vectors, which illustrate proteins and functional interactions, respectively. As we previously proved [131,132,133,134,135], this allows the delineation of affected/deregulated pathways and highly significant biomarkers characterizing the investigated biological state.

### 3.13. Statistical Analysis

Data are presented as mean ± SD of 3 experiments performed in triplicates and were analyzed using Graph Pad Prism 6–8 software. Groups were compared using *t*-tests. Statistical significance was set at *p* < 0.05.

## 4. Conclusions

In conclusion, we showed that, similarly to cholesterol loading, CSC induces SMC phenotypic switch by downregulating the miR145/MYOCD/KLF4 axis and by affecting the expression of diverse factors not previously associated with cellular response to cigarette smoke. By combining biological, transcriptional, proteomic, and bioinformatics resources we showed that CSC-deregulated factors are under a tight functional cross-talk between PKR and KLF4, hence suggesting that vascular SMC-transdifferentiation induced by CSC is orchestrated by PKR and KLF4. These functional cross-talks may offer a new perspective on cigarette smoke condensate effects on SMCs and they could be evaluated as biomarkers and targets in SMC phenotypic plasticity.

## Figures and Tables

**Figure 1 ijms-24-06431-f001:**
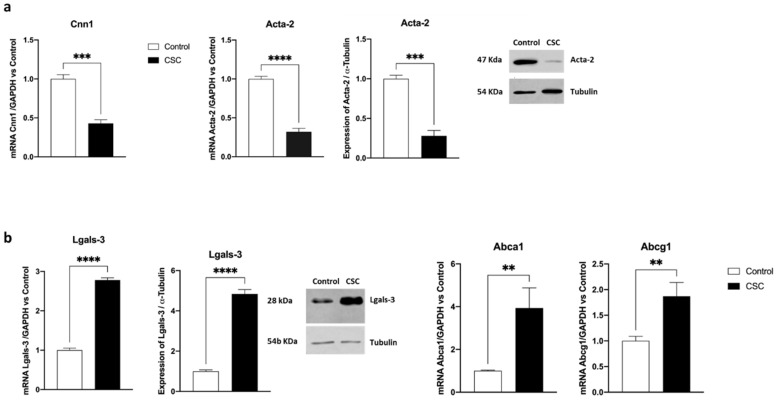
CSC induces a phenotypic switch in murine SMCs. Murine SMCs were incubated for 48 h with CSC (30 μg/mL). Then, the expression of contractile (**a**) or inflammatory (**b**) genes was evaluated using RT-PCR or WB analysis. Data are the mean ± SD of at least three experiments performed in triplicates. ** *p* < 0.01; *** *p* < 0.005; **** *p* < 0.001 vs. control.

**Figure 2 ijms-24-06431-f002:**
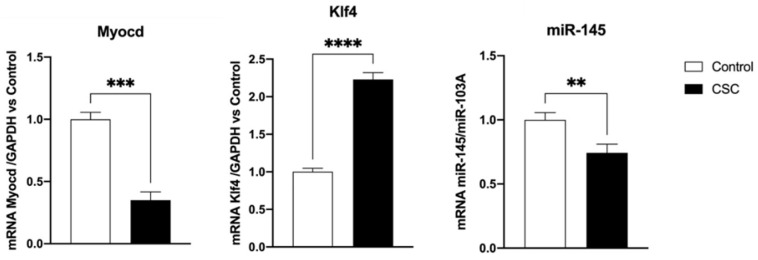
CSC affects the *Myocd/miR145/Klf4* axis in murine SMCs. Murine SMCs were incubated for 48 h with CSC (30 μg/mL). Then, the expression of *Myocd*, *Klf4*, and *miR-145* genes was evaluated using RT-PCR. Data are the mean ± SD of at least three experiments performed in triplicates. ** *p* < 0.01; *** *p* < 0.005; **** *p* < 0.001 vs. control.

**Figure 3 ijms-24-06431-f003:**
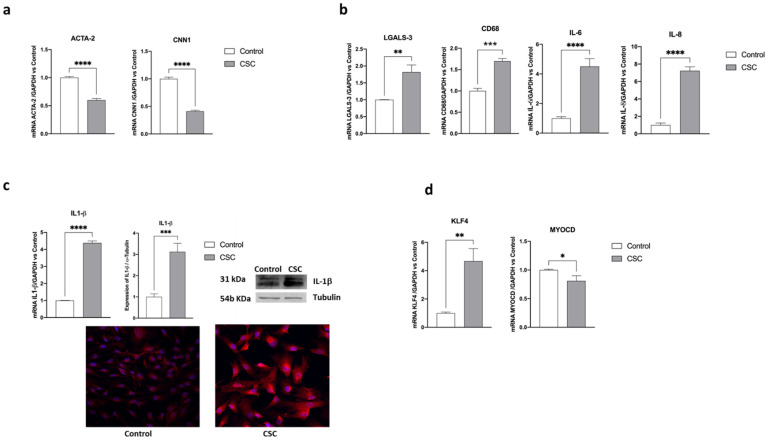
CSC induces HSMC phenotypic switch. Human SMCs were incubated for 48 h with CSC (30 μg/mL). Then, the expression of (**a**) contractile or (**b**,**c**) inflammatory genes was evaluated using RT-PCR. (**c**) IL-1β protein expression was evaluated using RT-PCR, western blot, and confocal microscopy analysis. Images were captured at 40× magnification using a FRET FLIM confocal microscope. (**d**) The expression of *KLF4* and *MYOCD* genes was measured using RT-PCR. Data are the mean ± SD of at least three experiments performed in triplicates. * *p* < 0.05, ** *p* < 0.01; *** *p* < 0.005; **** *p* < 0.001 vs. control.

**Figure 4 ijms-24-06431-f004:**
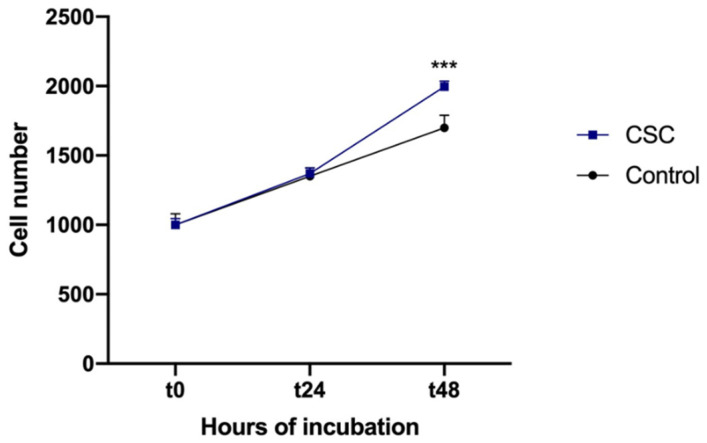
CSC stimulates HSMC proliferation. Human SMCs were incubated for 48 h with CSC (30 μg/mL). Then, cell proliferation was evaluated using cell counting. Data are the mean ± SD of at least four experiments performed in triplicates. *** *p* < 0.005 vs. control.

**Figure 5 ijms-24-06431-f005:**
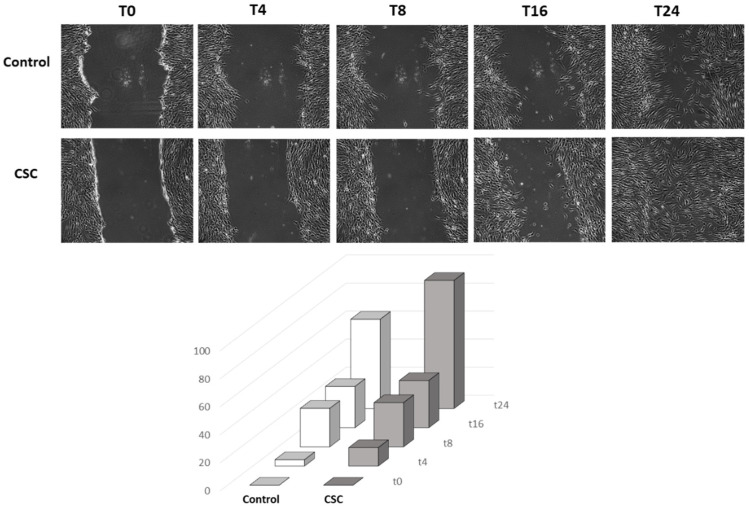
CSC stimulates HSMC migration. Human SMCs were incubated for 48 h with CSC (30 μg/mL). Then, cell migration was evaluated with the wound-healing assay. Images were captured at 10× magnification using an inverted microscope. Data are the mean ± SD of at least three experiments performed in triplicates.

**Figure 6 ijms-24-06431-f006:**
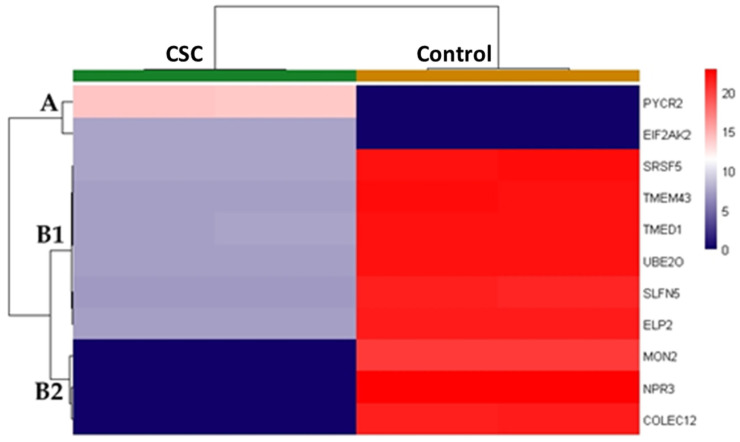
Heatmap of not-scaled Euclidean distances of abundance values from the 11 protein differences we identified between CSC-exposed (green bar in the horizontal dendrogram) and control (golden bar in the horizontal dendrogram) HSMCs by applying an MS-based shotgun proteomic approach.

**Figure 7 ijms-24-06431-f007:**
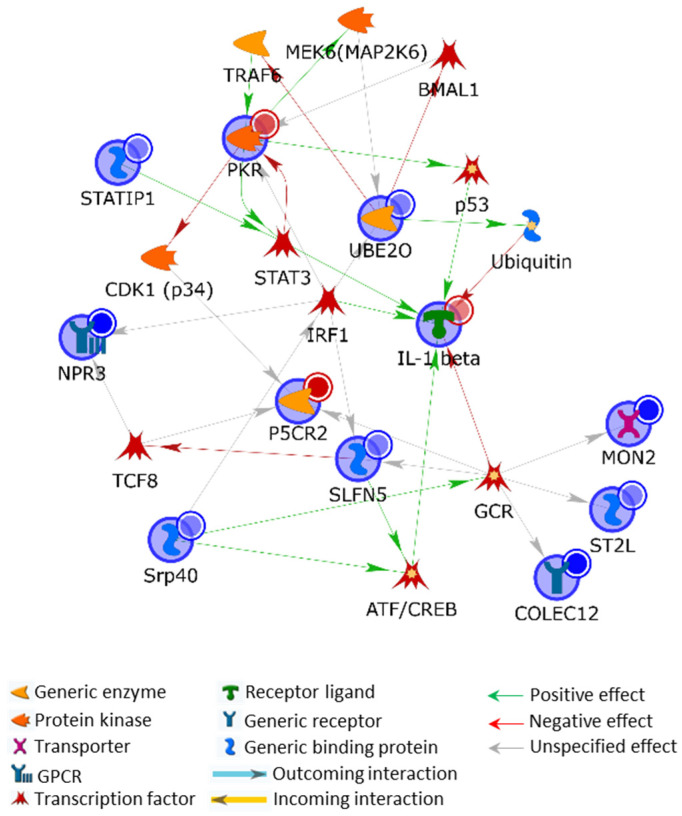
MetaCore SPN built by processing significant HSMC proteins deregulated by 48 h of CSC-treatment and identified using MS or detected using WB. Experimental factors, circled in blue, are cross-linked by expanding protein interactions to other factors, not present in the processed list but supported by the MetaCore database, that are needed to functionally correlate experimental protein differences that do not directly interact. Only TMEM43 did not enter into the SPN. Red and blue bubbles indicate the up- and downregulation, respectively, of the experimental factors in CSC-treated HSMCs.

**Figure 8 ijms-24-06431-f008:**
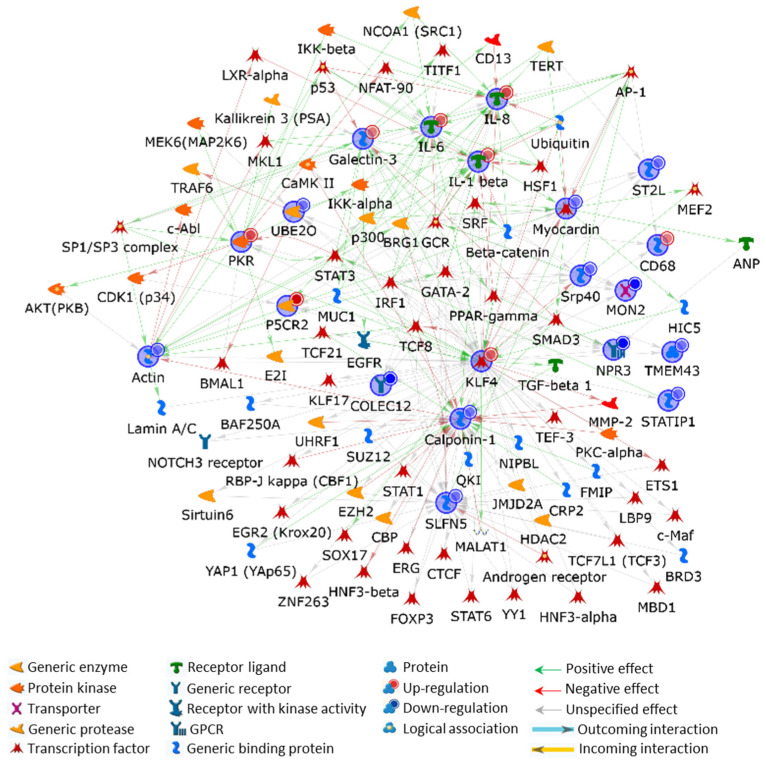
MetaCore SPN built by processing significant HSMC proteins and genes deregulated by 48 h of CSC treatment. Experimental factors, circled in blue, are cross-linked by expanding protein interactions to other factors, which are not present in the processed experimental-list but supported by the MetaCore database, and which are needed to functionally correlate user up-loaded proteins/genes that do not directly interact. All the processed deregulated proteins/genes were entered into the SPN. Red and blue bubbles indicate the up- or downregulation, respectively, of the experimental factors in CSC-treated HSMCs.

**Table 1 ijms-24-06431-t001:** Differentially abundant proteins, detected/identified using LC–MS/MS analysis, occurring between the CSC-treated and control human SMCs.

UniProtKB Protein Name	UniProtKB A.N.	Gene Symbol (Alternative Symbol)	CSC Abundance	Control Abundance	Coverage (Unique Peptides)	Score	Adj. *p* Value
Atrial natriuretic peptide receptor 3	P17342	***NPR3*** (*ANPRC*, *C5orf23*, *NPRC*)	0 (0)	22.8 (0.2)	28.1% (9)	22.7	8.08 × 10^−4^
Collectin-12	Q5KU26	***COLEC12*** (*CLP1*, *NSR2*, *SCARA4*, *SRCL*)	0 (0)	21.6 (0.2)	7.7% (6)	8.2	7.92 × 10^−4^
Elongator complex protein 2	Q6IA86	*ELP2* (***STATIP1***)	7.1 (0.0)	21.6 (0.0)	9.2% (5)	18.4	5.28 × 10^−4^
(E3-independent) E2 ubiquitin-conjugating enzyme	Q9C0C9	***UBE2O*** (*KIAA1734*)	7.3 (0.0)	22.1 (0.2)	7.3% (6)	20.5	1.17 × 10^−3^
Interferon-induced, double-stranded RNA-activated protein kinase	P19525	*EIF2AK2* (***PKR***, *PRKR*)	7.4 (0.0)	0 (0)	14.8% (7)	13.2	9.32 × 10^−4^
Protein MON2 homolog	Q7Z3U7	***MON2*** (*KIAA1040*, *SF21*)	0 (0)	20.2 (0.0)	2% (3)	5.0	5.04 × 10^−4^
Pyrroline-5-carboxylate reductase 2	Q96C36	** *PYCR2* **	14 (0.1)	0 (0)	15.9% (2)	5.0	8.45 × 10^−4^
Schlafen family member 5	Q08AF3	** *SLFN5* **	7 (0.0)	21.2 (0.2)	8% (5)	11.3	1.34 × 10^−3^
Serine/arginine-rich splicing factor 5	Q13243	*SRSF5* (*HRS*, *SFRS5*, ***SRP40***)	7.4 (0.1)	22.3 (0.2)	11.4% (2)	4.4	1.19 × 10^−3^
Transmembrane emp24 domain-containing protein 1	Q13445	*TMED1* (*IL1RL1L*, *IL1RL1LG*, ***ST2L***)	7.3 (0.0)	22.1 (0.1)	9.7% (2)	4.0	5.28 × 10^−4^
Transmembrane protein 43	Q9BTV4	** *TMEM43* **	7.3 (0.1)	22.1 (0.4)	22% (6)	20.2	4.64 × 10^−3^

The table reports, for each significant protein difference obtained using MS, the recommended UniProtKB protein name; UniPtotKB accession number; corresponding gene symbol (alternative symbol(s)) and, in bold, the MetaCore protein symbol; the mean LFQ-abundance (and standard deviation) in CSC and control groups; and the MS identification result in terms of % protein coverage (and number of identified unique peptides), score, and adjusted *p* value.

**Table 2 ijms-24-06431-t002:** Sequences of mouse primers.

Gene Name	Sequences	Gene Name	Sequences
*Abca1*	FW 5′-AAAACCGCAGACATCCTTCAG-3′ RV 5′-CATACCGAAACTCGTTCACCC-3′	*Klf4*	FW 5′-CTTTCCTGCCAGACCAGATG-3′ RV 5′-GGTTTCTCGCCTGTGTGAGT-3′
*Abcg1*	FW 5′-CCTTATCAATGGAATGCCCCG-3′ RV 5′-CTGCCTTCATCCTTCTCCTG-3′	*Lgals-3*	FW 5′-TGGGCACAGTGAAACCCAAC-3′ RV 5′-TCCTGCTTCGTGTTACACACA-3′
*Acta2*	FW 5′-GTCCCAGACATCAGGGAGTAA-3′ RV 5′-TCGGATACTTCAGCGTCAGGA-3′	*Myocd*	FW 5′-AAGGTCCATTCCAACTGCTC-3′ RV 5′-CCATCTCTACTGCTGTCATCC-3′
*Cnn1*	FW 5′-TTGAGAGAAGGCAGGAACATC-3′ RV 5′-GTACCCAGTTTGGGATCATAGAG-3′		

## Data Availability

MS data are available via ProteomeXchange with identifier PXD041174 (Project DOI: 10.6019/PXD041174).

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
