# Peer review of "Smooth Muscle Cell Phenotypic Switch Induced by Traditional Cigarette Smoke Condensate: A Holistic Overview"

_ijms, 2023, doi:10.3390/ijms24076431_

Round 1

Reviewer 1 Report

This manuscript describes a research in which cigarette smoke condensate’s (CSC) effects on atherosclerosis-related inflammatory activity were investigated on human and mouse smooth muscle cells (SMCs) in vitro. Specific cell markers, inflammatory factors, and signaling factors were tested, showing that CSC can promote the macrophage-directed metaplasia or trans-differentiation in both human and mouse SMCs, and stimulate SMCs migration. Overall, the contents in this work are abundant, and the data are logically arranged, which brings this study scientific soundness. The results are clear, bringing novelty to this study. I listed several major concerns need to be addressed.  

1.    The method design is confusing. One big question to make clear: what’s the role of CSC in the inflammatory pathway under SMCs’ switching, a regulator or stimulus? To test the inflammatory activity via IL-1β, IL-1β secretion assay should be done after the primary treatment of regulator (CSC for a longer time), then sequentially secondary treatment of stimulus (such as LPS for a shorter time). At last, ATP should be used to trigger the release of IL-1β after the primary and secondary treatments, and ELISA should be adopted to test the IL-1β level, not western blot. In the manuscript, the cells were treated with CSC only before the cells were lysed for IL-1β western blot. Stimulus was not used and IL-1β should be tested by ELISA rather than western blot. I guess the authors intended to check the upregulation of the inflammatory activity which was induced by CSC. If so you should do western blot for pro-IL-1β or ASC expression. I will show why below.

2.    The inflammatory activity in macrophages is mainly mediated by NLRP3 pathway, where stimulus like LPS binds toll-liker receptor (TLR) and triggers NF-κB as a transcription factor to upregulate pro-IL-1β expression. Meanwhile TLR stimulated the assembly of NLRP3 complex (ASC protein included) to activate caspase-1, which cleaves pro-IL-1β to IL-1β to be secreted extracellularly. That’s why IL-1β should be targeted outside macrophage-like cells not inside. This pathway and IL-1β secretion assay method were illustrated in previous publications (PMID: 34197316), Please explain and discuss more about this mechanism and method in the manuscript.

3.    It’s interesting to see the effect of CSC on IL-6, because many studies have shown the role of IL-6 in cardiovascular disease (PMID: 23906495, 35948913). It is of much value to talk about this point to show the effects of CSC on cardiovascular disease via the bridge of IL-6.

4.    The cell markers CD11b, CD11c and Ly6c have been shown specific for macrophage during the macrophage-directed differentiation (PMID: 35284603). They can serve better to show the identity shift during the trans-differentiation. For the readers’ benefit and interest, it is suggested to discuss more about this in the manuscript.

Author Response

We are submitting the revised version of the manuscript ijms-2249576 by Bianchi L. et al. We are grateful to Reviewers for their comments as the suggested changes have improved the clarity of the manuscript. This latter has been revised, when it was possible, based on the requests provided by the Reviewer #1 (Rev. #1) and Reviewer #3 (Rev. #3), as Reviewer #2 did not request any change.

A detail point-by-point replay is reported below. All changes in the manuscript have been marked up by using the “Track Changes” function, as suggested by the Assistant Editor, Ms. Dragana Danilov.

Rev. #1

This manuscript describes a research in which cigarette smoke condensate’s (CSC) effects on atherosclerosis-related inflammatory activity were investigated on human and mouse smooth muscle cells (SMCs) in vitro. Specific cell markers, inflammatory factors, and signaling factors were tested, showing that CSC can promote the macrophage-directed metaplasia or trans-differentiation in both human and mouse SMCs, and stimulate SMCs migration. Overall, the contents in this work are abundant, and the data are logically arranged, which brings this study scientific soundness. The results are clear, bringing novelty to this study. I listed several major concerns need to be addressed.  

  1. The method design is confusing. One big question to make clear: what’s the role of CSC in the inflammatory pathway under SMCs’ switching, a regulator or stimulus? To test the inflammatory activity via IL-1β, IL-1β secretion assay should be done after the primary treatment of regulator (CSC for a longer time), then sequentially secondary treatment of stimulus (such as LPS for a shorter time). At last, ATP should be used to trigger the release of IL-1β after the primary and secondary treatments, and ELISA should be adopted to test the IL-1β level, not western blot. In the manuscript, the cells were treated with CSC only before the cells were lysed for IL-1β western blot. Stimulus was not used and IL-1β should be tested by ELISA rather than western blot. I guess the authors intended to check the upregulation of the inflammatory activity which was induced by CSC. If so you should do western blot for pro-IL-1β or ASC expression. I will show why below.

We thank Rev. #1 for this interesting suggestion. We understand the subtle distinction that the Reviewer makes regarding the role of CSCs as regulator or a stimulus and we agree with her/him on the importance of evaluating extracellular IL-1β content as well. However, this was not the goal of our work.  First of all, we would like to point out that we did not conduct our analyses in cells “treated with CSC only before they were lysed for IL-1β western blot”, as indicated by Rev. #1. All the experiments were performed on cells exposed to CSC for 48 hours, as described in the” materials and methods” section: "Cells used in experiments were treated with CSC 30 ug/mL for 48 hs before evaluations, as previously shown [135]", lines 544-545, and in the “results and discussion” section: “Next, we confirmed the murine data in human cells. Human aortic SMCs (HSMCs) were incubated for 48 hours with CSC (30 µg/mL) and then HSMC phenotypic switch was evaluated”, lines 136-137.

Furthermore, the Ab we used (ab2105 by Abcam) to immunodetect IL-1β was obtained, according to the corresponding Abcam data sheet, by using the following immunogen “Recombinant full-length protein corresponding to Human IL-1 beta aa 100 to the C-terminus. Produced in E. coli. MW of recombinant IL-1 beta 17kDa, mature chain without propeptide. Database link: P01584”. Obviously, this Ab allows the detection of the precursor as well as of the mature chain of IL-1β. According to the MW of the two bands (at about 30 and 32 kDa), whose values were added (as 31 kDa mean value) in the reviewed Figure 3 for more clarity, that we detected and showed in Figure 3, the IL-1β we immunostained bona fide corresponds to two proteoforms of the pro-IL-1β, as named by Rev. #1. Based on the world leading protein data base, i.e. UniProtKB, IL-1β (P01584) is the recommended name used to indicate the precursor (269 aa in humans) as well as its mature chain (from the 117th amino acid to the C-terminus end). Consequently, we properly tested IL-1β (precursor and mature chain) and we correctly referred to it as IL-1β.

Then, we would like to stress that we analysed the levels of IL-1β synthesis in SMCs, exposed for 48 hours to CSC, merely to evaluate whether or not CSC treatment is associated with a change in the cellular inflammatory state. In fact, IL-1β synthesis increases when the cellular inflammatory state increases. IL-1β augmented presence was first predicted by RT-PCR and then confirmed by western blot (WB) and immunocytochemistry. As it was performed on protein extract from cell pellets, the WB analysis can be considered an estimation of intracellular IL-1β protein occurrence, as well as the immunocytochemistry. The latter was actually performed on permeabilized cells and, as clearly evident in Figure 3C, the distribution of the immunofluorescent signal delineates the cell bodies of the synthetising cells (see “subcellular localization” in UniProtKB P01584-entry). In any case, we did not need a quantification of the amount of IL-1β expressed/produced by the CSC-treated SMCs, but only evidence of the proinflammatory effects of CSC treatment. The obtained results confirm an increased IL-1β synthesis and its endocellular presence associated with CSC exposure, regardless of the CSC regulatory or stimulatory function.

In conclusion, we proved that CSC treatment correlates with an increased synthesis of one of the main relevant factors classically associated with the cellular inflammatory state, i.e. IL-1β, and this is what we desired to prove, independently on its secretion process.

  1. The inflammatory activity in macrophages is mainly mediated by NLRP3 pathway, where stimulus like LPS binds toll-liker receptor (TLR) and triggers NF-κB as a transcription factor to upregulate pro-IL-1β Meanwhile TLR stimulated the assembly of NLRP3 complex (ASC protein included) to activate caspase-1, which cleaves pro-IL-1β to IL-1β to be secreted extracellularly. That’s why IL-1β should be targeted outside macrophage-like cells not inside. This pathway and IL-1β secretion assay method were illustrated in previous publications (PMID: 34197316), Please explain and discuss more about this mechanism and method in the manuscript.

Once again, we understand the interest of Rev. #1 in the mechanisms of IL-1β secretion. We reiterate that this is not the target of our work. Also, on the basis of our answer to the Rev. #1’s comment no. 1, we sustain that the evaluation of the increased IL-1β synthesis is sufficient to demonstrate that the 48 hours exposure to CSC correlates with an increased cellular inflammatory state. The signalling pathways leading to this increase, or whether or not CSC increases IL-1β secretion in SMCs exhibiting phenotypic variations, cannot be investigated in this work. This evaluation would require a series of experiments and data discussion that would not only burden the work but, above all, would divert attention from the other series of relevant data we have dealt with, first of all the identification of factors not classically associated with SMC transdifferentiation and their predictive functional correlation. For example, the IL-1β secretion mechanism related to CSC-treatment in SMCs during their phenotypic switch, and not in macrophages o macrophage-like cells, may follow different, probably unconventional, or parallel pathways not yet properly described, as we suggest in the results and discussion section: “In fact, PKR is induced by pro-inflammatory cytokines, e.g. TNF-α, IL-1, INF-γ, and, depending on cell type, PKR itself induces the release of the pro-inflammatory IL-18, IL-1β, and high mobility group box 1 (HMGB1) alarmin proteins [40]”, lines 271-274. For this reason, we decided not to comment on the signalling and secretion pathways suggested by the Reviewer. In fact, they are typical of bone marrow-derived macrophages, a very different condition from the one we evaluated in our work.

In conclusion, we are very sorry that Rev. #1 focused mainly on the aspect of IL-1β secretion and on the mechanisms of its activation in macrophages, not evaluating the consistent work of differential proteomics and the interpretation of the data obtained through a complex functional predictive assessment of all the deregulated factors identified with complementary experimental methods. Functional transcriptomics/proteomics are the main applied processes to obtain a holistic view on the system under analysis, and this was what we tried to obtain evaluating CSC effects in SMCs.

  1. It’s interesting to see the effect of CSC on IL-6, because many studies have shown the role of IL-6 in cardiovascular disease (PMID: 23906495, 35948913). It is of much value to talk about this point to show the effects of CSC on cardiovascular disease via the bridge of IL-6.

We thank again the Rev.#1 for this comment. We are very well aware that IL-6 is involved in cardiovascular disease and that it is associated with the genetic prediction of cardiovascular disease risk and we added the corresponding reference in the text (lines 307-309). However, IL-6 pathway is driven by IL-1β which is central in the inflammatory response and is very much associated with cardiovascular complications. In fact, an anti-inflammatory therapy targeting the IL-1β innate immunity pathway with canakinumab led to a significantly lower rate of recurrent cardiovascular events, independent of lipid-level lowering (PMID: 28845751, Cantos trial). Therefore, we feel that our data on CSC effects on IL-1β shows and confirms quite well the involvement of cigarette smoke in cardiovascular diseases.

  1. The cell markers CD11b, CD11c and Ly6c have been shown specific for macrophage during the macrophage-directed differentiation (PMID: 35284603). They can serve better to show the identity shift during the trans-differentiation. For the readers’ benefit and interest, it is suggested to discuss more about this in the manuscript.

We agree with Rev. #1 on the importance of CD11b, CD11c, and Ly6c for macrophage-directed differentiation by genetic manipulation of mouse embryonic stem cells. Once again we would like to underline that, in our work, we describe a phenotypic change of SMCs towards a probable state of mesenchymal-like phenotype from which, according to the literature, the macrophage-like one originates. The two biological contexts, the one suggested by the Reviewer and the one investigated by us, are therefore different. In our work we tested, among the markers of the inflammatory phenotype, also CD68 whose CSC-induced up-regulation could facilitate the subsequent differentiation of mesenchymal-like SMCs to macrophage-like cells, as discussed in the results and discussion section: “As CSC induces CD68 expression in SMCs in vivo, we can suppose that the lipophilic components of CSC not only induce vascular SMC switch to the mesenchymal-like phenotype but that it may even facilitate its further switch to the macrophage-like one”, lines 512-515. Differential proteomic analysis and predictive functional analysis did not reveal concentration differences of CD11b, CD11c and Ly6c or their inclusion, as non-experimental factors in the generated networks. Therefore, we think that there is no particular reason to comment also these proteins, further burdening our work.

In conclusion, the identity change mentioned by the Reviewer could occur, during CSC induction, by pathways at least partially different from those associated with the classic SMC stressors that induce the onset and development of atheromatous plaques. Consequently, we feel that commenting on factors not highlighted by our analyses would be misleading for the reader.

Reviewer 2 Report

Vascular smooth muscle cells play an important role in the pathogenesis of atherosclerosis. Understanding these pathophysiological mechanisms will improve the efficiency of diagnosis and treatment of atherosclerosis. The data obtained by the authors are of research and clinical interest. The data obtained by the authors were confirmed by a sufficient number of informative figures and an extensive discussion, which is also of interest

Author Response

We are submitting the revised version of the manuscript ijms-2249576 by Bianchi L. et al. We are grateful to Reviewers for their comments as the suggested changes have improved the clarity of the manuscript. This latter has been revised, when it was possible, based on the requests provided by the Reviewer #1 (Rev. #1) and Reviewer #3 (Rev. #3), as Reviewer #2 did not request any change.

A detail point-by-point replay is reported below. All changes in the manuscript have been marked up by using the “Track Changes” function, as suggested by the Assistant Editor, Ms. Dragana Danilov.

Rev. #2

Vascular smooth muscle cells play an important role in the pathogenesis of atherosclerosis. Understanding these pathophysiological mechanisms will improve the efficiency of diagnosis and treatment of atherosclerosis. The data obtained by the authors are of research and clinical interest. The data obtained by the authors were confirmed by a sufficient number of informative figures and an extensive discussion, which is also of interest.

We are very grateful to Rev. #2 for her/his appreciation of our work.

Reviewer 3 Report

Bianchi et al. examined the role of cigarette smoke condensate (CSC) in the gene expression of smooth muscle cells.  By using RT-PCR, western blotting, and MS-based differential proteomic approach, they showed that CSC induced phenotypic switching in SMCs in vitro.  The authors should address the following points.

Comments:

1. The authors concluded that PKR and KLF4 are key factors in phenotypic switching of SMCs by CSC.  Unfortunately, they did not perform any loss-of-function experiments.  Use pharmacological inhibitors, siRNAs, or gene knockout cells to test if loss of PKR affects CSC-induced SMC phenotypic switching. 

2. Results and Discussion have been written together.  Please describe these sections separately. 

3. Discussion is too long and so speculative. 

4. line 20: MS-based, line 27: PKR; Do not use abbreviations without definition. 

5. line 51: CNN1, line 58: calponin; Once defined, please use abbreviated forms subsequently. 

6. Several sentences are difficult to understand.  For example, line 115-117. 

7. Line 203-211: This paragraph has been written for the next topic (protein hybrid network). 

Author Response

We are submitting the revised version of the manuscript ijms-2249576 by Bianchi L. et al. We are grateful to Reviewers for their comments as the suggested changes have improved the clarity of the manuscript. This latter has been revised, when it was possible, based on the requests provided by the Reviewer #1 (Rev. #1) and Reviewer #3 (Rev. #3), as Reviewer #2 did not request any change.

A detail point-by-point replay is reported below. All changes in the manuscript have been marked up by using the “Track Changes” function, as suggested by the Assistant Editor, Ms. Dragana Danilov.

Rev. #3

Bianchi et al. examined the role of cigarette smoke condensate (CSC) in the gene expression of smooth muscle cells.  By using RT-PCR, western blotting, and MS-based differential proteomic approach, they showed that CSC induced phenotypic switching in SMCs in vitro.  The authors should address the following points.

 Comments:

  1. The authors concluded that PKR and KLF4 are key factors in phenotypic switching of SMCs by CSC. Unfortunately, they did not perform any loss-of-function experiments. Use pharmacological inhibitors, siRNAs, or gene knockout cells to test if loss of PKR affects CSC-induced SMC phenotypic switching.

We thank the Reviewer for his valuable suggestion which we will certainly take into consideration for future analyses we have already planned. However, what Rev #3 suggests does not fit with the main purpose of this work, which was just to demonstrate that CSC induces SMC phenotypic modulation. Our analysis also predictively demonstrates that this process involves different factors whose functions are modulated by two key proteins: KLF4 and PKR. And that is what we want to communicate. Obviously, further investigations are necessary to corroborate the data we obtained and to demonstrate how those factors operate in the investigated biological-context, which even involves other factors not previously associated to the “classical” mesenchymal-like phenotype. These investigations would involve several experiments and the presentation and discussion of a lot of other new data, and we will be able to do this properly only in another work.

  1. Results and Discussion have been written together. Please describe these sections separately.

Frequently, results from predictive functional analysis that leads to the generation of networks are presented together with their discussion to facilitate their treatment and for better understanding of the text. In this way, possible repetitions are avoided in the discussion and the flow of the dissertation is more fluid. Networks are often not sufficiently analysed and discussed in papers because their treatment is complex and occupies a large part of the discussion. In our manuscript, we decided to focalize our attention and to use a large part of the results and discussion section to exhaustively evaluate the systemic functional significance of the deregulated factors entered into the nets and the cross-talks established among them, and also among them and the non experimental proteins added by the software for their reciprocal linking. For this reason, we opted to combine the results with their discussion. In addition, IJMS allows the merging of these sections, as attested by some of our previous work published in this Journal, e.g. PMID: 33406681 and 36362324.

We hope that Rev. #3 will agree with us, and taking into account that the other two Reviewers did not highlight this problem, we would be grateful, for the clarity of the text, if we are allowed to keep the results together with their discussion.

  1. Discussion is too long and so speculative.

We are very sorry that Rev. #3 considers the description of the network and the discussion of the functions and relevance of its factors mere speculation. Even more since another Reviewer actually appreciated it. We have tried to reduce the discussion, where it was possible without affecting the significance of the manuscript, to lighten this section as suggested by the Reviewer. However, functional bioinformatic analyses are predictive, but this does not imply that they do not reflect a real biological situation or one that is very close to reality. Our experience in generating networks allows us to affirm that it is almost impossible to obtain statistically significant networks starting from a few experimental factors if these factors do not present real interactions described in the literature. The achievement of a highly significant network with our data underscores that the factors we found to be deregulated play key roles in the mechanisms leading to CSC-induced SMC phenotypic change. These factors have already been described in the literature, some in mutual correlation, in inflammatory, fibrotic, apoptotic, oxidative stress, and aberrant metabolic conditions, all processes associated with atherosclerotic phenomena. Therefore, we feel that a detailed discussion of the mechanisms in which they may be implied and their reciprocal correlations, despite this takes a large part of the discussion, is necessary to underline how their functional and systemic combination can correlate with the well-known increased atherosclerotic risk in smokers. Furthermore, digressions on non-experimentally identified proteins added by the software to cross-link experimental deregulated factors, e.g. the signal transducer activator of transcription 3 (STAT3) and the interferon regulatory factor 1 (IRF1), are of relevance and are classically present in network discussions. In fact, these proteins often corroborate the functional meaning of experimental factors by correlating them with known pathways and processes. In addition, added proteins are often taken into consideration for successive experimental analyses to improve the comprehension of biochemical mechanisms, supported by the affected factors, that lead to phenotypic modulation. What for Rev. #3 is speculation, for us is explanation and discussion of the network results.

  1. line 20: MS-based, line 27: PKR; Do not use abbreviations without definition.

We apologize for the inconvenience; we have corrected the text according to the suggestion of the Reviewer. 

  1. line 51: CNN1, line 58: calponin; Once defined, please use abbreviated forms subsequently.

We apologize for the inconvenience; we have corrected the text according to the suggestion of the Reviewer.

  1. Several sentences are difficult to understand. For example, line 115-117.

In the “Quality of English Language”, Rev. #3 selected: “English language and style are fine/minor spell check required”, as both Rev. #1 and Rev. #2 also did. As a consequence, it is difficult for us to identify sentences that need to be improved. We would be very grateful to Rev. #3 if she/he could tell us which sentences need English improvement so that we can increase their clarity, as we did for lines 115-117 (now 122-126).

  1. Line 203-211: This paragraph has been written for the next topic (protein hybrid network).

This was done on purpose to introduce the next two sub-paragraphs of the paragraph “2.3. CSC effects on VSMC proteomic profile and combined functional analyses of proteins and genes significantly deregulated by the CSC treatment”. Actually, while the first part of the paragraph reports relevant information, i.e. MS data, for both hybrid networks, the following sub-paragraphs describe the two different hybrid networks we generated and discuss their functional (predictive) relevance. Line 203-211 (now lines 215-223) are needed to create continuity with the previous part of 2.3 paragraph and to introduce how we processed the proteins identified by MS.

Round 2

Reviewer 1 Report

This manuscript describes a research in which cigarette smoke condensate’s (CSC) effects on atherosclerosis-related inflammatory activity were investigated on human and mouse smooth muscle cells (SMCs) in vitro. Specific cell markers, inflammatory factors, and signaling factors were tested, showing that CSC can promote the macrophage-directed metaplasia or trans-differentiation in both human and mouse SMCs, and stimulate SMCs migration. Overall, the contents in this work are abundant, and the data are logically arranged, which brings this study scientific soundness. The results are clear, bringing novelty to this study. The authors responded well to my questions and made detailed revisions. After explanations, the conclusion is convincing.

Reviewer 3 Report

The manuscript has been improved.